# Parkinson’s Disease, It Takes Guts: The Correlation between Intestinal Microbiome and Cytokine Network with Neurodegeneration

**DOI:** 10.3390/biology12010093

**Published:** 2023-01-07

**Authors:** Georgia Xiromerisiou, Chrysoula Marogianni, Anastasia Androutsopoulou, Panagiotis Ntavaroukas, Dimitrios Mysiris, Stamatia Papoutsopoulou

**Affiliations:** 1Department of Medicine, Faculty of Life Sciences, University of Thessaly, 41500 Larisa, Greece; 2Department of Biochemistry and Biotechnology, Faculty of Life Sciences, University of Thessaly, 41500 Larisa, Greece

**Keywords:** Parkinson’s disease, cytokines, chemokines, gut, microbiota

## Abstract

**Simple Summary:**

Parkinson’s disease is a neurodegenerative disease of the central nervous system, characterized by movement problems and accompanied by behavioral changes such as depression and anxiety. It is a multifactorial condition that is affected by genetic alterations and environmental factors that progressively lead to the death of specialized neurons. This systematic review discusses the attractive hypothesis that gut intestinal dysbiosis is an initial step of a process that leads to Parkinson’s. Gut microbiota alterations and their metabolites can cause intestinal inflammation, but they can also alter gut-brain communication and the brain barrier. This can lead to brain inflammation and the deterioration of brain cells, a process called neurodegeneration. Understanding the role of gut microbiota in the progression of Parkinson’s could be a key element for research toward therapeutic approaches that could delay and even cure the disease.

**Abstract:**

Parkinson’s disease is a progressive neurodegenerative disorder with motor, physical and behavioral symptoms that can have a profound impact on the patient’s quality of life. Most cases are idiopathic, and the exact mechanism of the disease’s cause is unknown. The current hypothesis focuses on the gut-brain axis and states that gut microbiota dysbiosis can trigger inflammation and advances the development of Parkinson’s disease. This systematic review presents the current knowledge of gut microbiota analysis and inflammation based on selected studies on Parkinson’s patients and experimental animal models. Changes in gut microbiota correlate with Parkinson’s disease, but only a few studies have considered inflammatory modulators as important triggers of the disease. Nevertheless, it is evident that proinflammatory cytokines and chemokines are induced in the gut, the circulation, and the brain before the development of the disease’s neurological symptoms and exacerbate the disease. Increased levels of tumor necrosis factor, interleukin-1β, interleukin-6, interleukin-17A and interferon-γ can correlate with altered gut microbiota. Instead, treatment of gut dysbiosis is accompanied by reduced levels of inflammatory mediators in specific tissues, such as the colon, brain and serum and/or cerebrospinal fluid. Deciphering the role of the immune responses and the mechanisms of the PD-associated gut microbiota will assist the interpretation of the pathogenesis of Parkinson’s and will elucidate appropriate therapeutic strategies.

## 1. Introduction

Parkinson’s disease (PD) is a slow-progressing neurodegenerative disease with high prevalence, currently affecting over 6 million people worldwide [1]. PD is mainly characterized by the impairment of motor function, as well as by a variety of nonmotor manifestations, including hyposmia, dysautonomia (e.g., orthostatic hypotension.), neurocognitive impairment, and sleep disturbances, all of which can have detrimental impacts on the patient’s quality of life [2,3,4]. There are currently no therapeutic approaches that can delay the progression of PD, only symptomatic treatments [5,6]. Sporadic forms of PD are thought to result from complex interactions between different genetic and environmental factors [7]. Up to 15% of patients with PD have genetic alterations [8]. Although these genetic variants frequently result in familial PD, illness penetrance varies greatly across mutation carriers, indicating that additional genetic modifiers or non-genetic environmental factors may also have an impact on the development of the disease. From a pathological perspective, PD is mainly characterized by alpha-synuclein misfolding, which results in the amyloid formation of insoluble aggregates (i.e., Lewy bodies and neuritis) in neurons, as well as glia, followed by subsequent neurodegeneration [8,9,10]. There is increasing evidence that gut-brain communication contributes to the development and progression of PD and involves the role of a-synuclein signaling [11,12,13]. Gut microbiota is considered an important mediator in communication that influences brain development and function [14]. In healthy individuals, the crosstalk between gut microbiota and their host is typically mutually beneficial. However, the fact that environmental factors impact the structure and function of microbiota in the alteration of their numbers and/or species could advance the development of certain diseases, a state known as dysbiosis. [15,16,17,18]. Many studies support that gut dysbiosis, which leads to a leaky gut, an alteration in the blood–brain barrier and neuroinflammation, is prevalent in PD [19,20,21,22,23,24,25]. Additionally, there is experimental evidence that gut microbiota induces a-synuclein synthesis [26]. The first study states that gut lumen (*Escherichia coli*) releases the extracellular amyloid protein Curli [27], which in rats could induce the disposition of a-synuclein in both the gut and the brain [28]. Subsequent studies have revealed that the presence of Curli in the gut can modulate a-synuclein aggregation [29,30,31]. Similarly, the lipopolysaccharide (LPS) expressed on the plasma membrane of Gram-negative bacteria has been shown to induce a-synuclein aggregation and provides another explanation of how microbiota contributes to PD [32,33,34,35]. Therefore, the role of gut microbiota in PD represents a significant area of research as they are potential therapeutic targets for the development of new strategies to tackle this disease [36].

Intestinal microbiota is also known to induce inflammation in conditions like dysbiosis, a topic that has been discussed extensively in other studies but not in relation to PD [37]. Studies on both human PD and animal models note that both innate and adaptive immune responses are affected during the disease [38,39]. Research on human donors has focused on peripheral inflammation, mainly due to the nature of the disease and the availability of samples. Increased levels of proinflammatory cytokines and chemokines, such as interleukin-6 (IL-6), tumor necrosis factor (TNF), interleukin-1β (IL-1β), chemokine (C-C motif) ligand 5 (CCL5) and interleukin-2 (IL-2) have been detected in serum, cerebrospinal fluid (CSF) and the brains of PD patients [40,41,42,43,44]. A recent hypothesis indicates that PD-related peripheral inflammation is a result of intestinal barrier deterioration, which causes systemic exposure to bacterial products such as LPS [45]. Increased levels of LPS and decreased LPS-binding protein (LBP) in the blood and plasma of PD patients further support this hypothesis [46,47]. The role of LPS in the progression of PD led to the development of various animal experimental models, through which mechanistic and drug discovery studies related to the disease will be performed [46].

This review highlights the role of gut microbiota dysbiosis and consequent inflammation in the contribution of the gut-brain axis in Parkinson’s disease. We will discuss the current knowledge of gut microbiota and inflammatory mediators, cytokines and chemokines, as well as their contribution to the advancement of PD. The usage of experimental models and the application of advanced methodologies to human PD research will facilitate our understanding of the interplay between these factors. This will further help us to design strategies that will prevent peripheral inflammation and neuroinflammation, therefore inhibiting neurodegeneration in PD patients.

## 2. Materials and Methods

### 2.1. Information Sources and Search Strategy

In this systematic review, studies on the association of gut microbiota, inflammatory mediators and PD were searched in PubMed from inception to June 2022. The search terms were as follows: (Parkinson OR Parkinson’s disease OR Parkinson’s syndrome) AND (gut intestine flora) OR (microbiota) OR (microbiome) OR (flora) OR (gut microflora) OR (appendix) OR (vermiform appendix). References in the articles were assessed to retrieve additional potentially relevant studies. There were no language restrictions. The PRISMA guidelines were followed for this review [48], and the flow diagram is shown in Figure 1.

### 2.2. Assessment for Eligibility

Studies were included if they met the following criteria: (a) studies that had a cross-sectional, cohort, or case–control design; (b) studies that recruited subjects who met the PD diagnostic criteria; (c) studies in which microbiota was studied with the following tests: the GBT (Glucose Hydrogen Breath Test), LBT (Lactulose Hydrogen Breath Test) or JAC (Jejunal Aspirated Culture); (d) studies that compared the association of PD and microbiota; (e) animal studies; (f) studies with full texts available; and (g) studies that included measurement of cytokines and chemokines. The exclusion criteria were as follows: case reports, review articles, and letters. The manuscripts were assessed in duplicates by a team (AA, PD, SP).

## 3. Results

A total of 751 potentially eligible articles were identified based on the described search strategy. Of all the extracted articles, 382 were excluded because they were duplicates, reviews, meta-analyses, or irrelevant studies. Finally, 369 studies were screened again, and we included in the analysis 91 case series, case–control studies and clinical trials and 77 animal studies. In total, four out of the human studies and 33 out of the animal studies had included approaches for cytokine and chemokine measurement.

### 3.1. Metagenomics

PD has received a great deal of research attention in the field of metagenomics and the role of gut microbiota in the pathogenesis of PD [49]. The first metagenomic investigation was published by Scheperjans et al. in 2015, and it revealed a decline in Prevotellaceae in feces from PD patients compared to healthy donors, as well as a positive link between Enterobacteriaceae abundance and the motor phenotype, specifically postural instability, and gait deviations [50]. A year later, another study discovered changes in these bacterial families, where Enterobacteriaceae were more prevalent than Prevotellaceae [51]. It is noteworthy to mention that another study showed a correlation between increased intestinal permeability and the presence of E. coli, a bacterial species that is assigned to the family Enterobacteriaceae [52]. Also, the phylum Bacteriodetes was significantly reduced in the feces of patients with PD [51]. The same PD fecal samples had lower levels of short chain fatty acids (SCFAs), e.g., acetate, propionate and butyrate, an important mediator for the maintenance of a healthy intestinal barrier [51,53]. In a recent review, Castillo-Alvarez and Marzo-Sola discuss studies by research groups that demonstrated a relationship between PD and specific gut microbiota compositions [54]. These results show that some microbes that produce SCFAs that have anti-inflammatory effects are more prevalent in feces from control donors than in patients with PD, while other microorganisms with a more proinflammatory profile are more prevalent in patients with PD. The gut microbiome has been reported to correlate with the disease symptoms since the first metagenomic study; Barichella et al. also showed that differences in taxa abundances were associated with the disease duration and affected the clinical profile of the disease [55]. A metagenomic shotgun analysis, where fecal microbiomes of early stage, L-DOPA-naïve PD patients compared to healthy controls, revealed differences in colonic microbiota with the total virus abundance decreased in PD participants [56]. A brief summary of gut microbiota dysbiosis’s impact on gut and brain function and communication is presented in Figure 2.

### 3.2. Cytokines and Chemokines

It has been recently recognized that there is a low-grade inflammation in the gut in PD. Gene expression studies showed upregulation of proinflammatory cytokines and chemokines in gut tissue from PD donors compared to healthy controls [57,58]. Specifically, Perez-Pardo et al. performed microarray analysis on human colonic biopsies and observed higher pro-inflammatory milieu in the tissues of PD patients, including, among others, upregulation of the *IFNG*, *IL1B*, *IL17A* and *IL8* genes [57]. In the same study, the authors clearly demonstrated the contribution of Toll-like receptor 4 (TLR-4) signaling in gut inflammation by showing that the TLR4 knockout mice were resistant to many of the PD-like consequences of rotenone-induced phenotype. Similarly, Devos et al. observed elevated mRNA expression levels of *TNF*, *IFNG*, *IL6* and *IL1B* in the ascending colon of PD patients [58]. Higher levels of IL-1α, IL-1β, IL-8 and C-reactive protein (CRP) have been found in stool homogenates from PD patients using a multiplexed immunoassay [59]. According to this study, the differences in those inflammatory mediators were not dependent on subject age or disease duration. Finally, increased numbers of T cells have been identified in colon tissue from PD patients compared to healthy individuals [57]. These findings support the hypothesis that gut inflammation could impact PD pathophysiology. In the following sections, the findings from human and experimental animal studies will be discussed separately.

#### 3.2.1. Human Studies

Only four of the selected studies had a combination of data about gut microbiota and cytokine or chemokine levels in PD patients. Lin et al. measured fecal microbiota in healthy and PD donors, as well as inflammatory mediators in serum, combining two different cohorts [60]. Among all the cytokines, only IFNγ and TNF plasma levels correlated with the altered gut microbiota. In a clinical trial in which samples were taken before and after treatment, it was found that berberine hydrochloride reduced both gut dysbiosis and serum levels of IL-6, IL-1β and TNF [61]. In a report by Aho et al., stool SCFAs and stool and plasma inflammatory mediators were measured and analyzed in relation to the disease (healthy versus PD donors) and gut microbiota [62]. Samples from PD patients had reduced SCFAs and showed signs of inflammation in correlation to the microbiota and disease onset, but they were not reflected in the systemic inflammatory profile. Lastly, a very recent study showed the beneficial effect of probiotics on PD patients using the M-SHIME^®^ system, a short-term culture of human fecal samples [48,63,64]. The results revealed changes in the microbial community, in specific SCFAs levels (acetate, propionate and butyrate) and lactate, which is the product of carbohydrate fermentation and key to good health. Moreover, colonic extracts from healthy and PD donors were tested in an in vitro setting and induced various cytokines and chemokines to a different degree [64].

#### 3.2.2. Experimental Animal Studies

Most of the animal studies were based on the MPTP (1-methyl-4-phenyl-1,2,3,6-tetrahydropyridine) model, which was established almost 40 years ago [65], followed by 6-OHDA (6-hydroxydopamine) [66] and rotenone-induced models [67]. Much fewer studies involved transgenic mice, such as the human α-synuclein overexpressing models [68,69]. The experimental approaches and the outcomes of the cytokine/chemokine measurements are described in Table 1. It is clear that gut microbiota dysbiosis correlates with inflammation in the gut and brain, as well as in the periphery, as reflected by plasma and serum data. Among the known inflammatory cytokines, there are differences in TNF, IL-6, IL-1β, IFN-γ and IL-17A, and in some cases, in the anti-inflammatory cytokine IL-10, which can be translated as a host response to dampen the inflammatory environment. Among the approaches to prevent the progress of PD and reduce tissue inflammation, one can see the use of antibiotics. In such a study, Koutzoumis et al. showed that 6-hydroxydopamine (6-OHDA)-induced neurotoxicity in rats was ameliorated by chronic treatment with a cocktail of broad-spectrum antibiotics (neomycin, vancomycin, bacitracin and pimarcin) [70]. The treatment led to a reduction of TNF and IL-1β levels in the striatum, as observed by real-time PCR. Also, oral administration of probiotics such as Lactobacillus plantarum reshaped the gut microbiota and caused an increase of the anti-inflammatory cytokine IL-10 and reduction of the proinflammatory cytokines TNF, IL-6 and IL-1β in a 1-methyl-4-phenyl-1,2,3,6-tetrahydropyridine (MPTP)-induced PD model in mice [71]. The use of TNF inhibitors in an MPTP-treated rhesus macaque monkey model showed that blocking of soluble TNF may have been associated with attenuated inflammation in biofluids, such as serum and CSF [72]. Lastly, fecal microbiota transplantation (FMT) has been utilized as a therapeutic approach in experimental animal models, such as an MPTP-induced mouse model [73]. In that study, Sun et al. showed that FMT not only affected the gut microbiota but had a remarkable effect on the brain, including a reduction of activation of microglia and astrocytes and TNF levels in the substantia nigra [73]. Similar studies using the FMT approach are presented in Table 1 below.

## 4. Discussion

This review discusses current findings regarding the contribution of gut microbiota dysbiosis and inflammation to Parkinson’s disease. Experimental animal studies have evidenced that PD is presented with gut dysbiosis and correlates with abnormal levels of inflammatory mediators in the periphery and central nervous system. In recent years, much research activity has been focused on the gut-brain axis, the bidirectional network between the gut and the brain, which involves the vagus nerve, immune factors, neuroendocrine pathways, and microbial metabolites [100]. The fact that PD is the most prevalent movement disorder and the second most prevalent neurodegenerative disease after Alzheimer’s has inevitably attracted much attention. An increasing number of studies feature observations that support the hypothesis that the pathogenesis of PD derives from the gut and that there is a correlation between PD and gastrointestinal diseases [101]. For example, one study observes Lewy pathology in the enteric nervous system before the onset of motor symptoms [102,103]. Additionally, some human studies demonstrate that the microbial populations in the gut of PD patients differ from the ones of healthy individuals [104,105,106]. Moreover, a metatranscriptomic analysis of the appendix microbiome, which is a lymphoid tissue that has been linked to a risk for PD development [107], reveals a microbial dysbiosis in PD patients, including an upregulation of bacteria that is responsible for secondary bile acid synthesis [108]. Changes in microbial populations inevitably affect the type and number of microbial metabolites, such as SCFAs, which play an important role in gut physiology and gut-brain communication [109,110,111]. A recent study exhibits a reduction in fecal SCFAs, but an increase in plasma SCFAs is observed in patients with PD, which correlates with specific gut microbiota changes and the clinical severity of PD [112]. Importantly, the gut microenvironment may be an element that modifies how PD symptoms develop in hereditary variants. Numerous rodent models of PD have been used to explore the intricate interactions between genetic risk factors in the host and environmental variables, particularly concerning gut microbes and gut metabolites. Genome-wide association studies (GWAS) have identified numerous genetic loci that increase their susceptibility to sporadic PD [113]. Among them, they have identified a number of genes linked to intestinal inflammation and gut microbial regulation, including *TLR1*, *TLR2*, *TLR4* and *MUC2*, the gene that encodes a component of the mucosal layer that protects the intestinal epithelial barrier [112]. The risk of inflammatory bowel disease (IBD) increases by some identified genetic factors that cause sporadic PD. For instance, *NOD2* is recognized to be a powerful indicator of IBD and to interact with *LRRK2* [114,115,116]. Gut dysbiosis has been shown to induce gastrointestinal inflammation and increase proinflammatory cytokines, including TNF, IFN-γ, IL-6 and IL-1β [116]. Since the first report by Devos et al. [58] that linked intestinal inflammation and PD, chronic gut inflammation and impaired intestinal barrier integrity have been observed in human PD patients and mouse models of PD [117]. Figure 2 summarizes the effect of gut microbiota dysbiosis, the induction of local and systemic inflammation, and the consequences that lead to neuroinflammation and PD.

There are several limitations in studies related to neurodegenerative diseases like PD. Firstly, the available experimental animal models are characterized by inconsistencies and, in some cases, the lack of Lewy body inclusions [118]. Yet, knowledge of the interplay between gut microbiota and intestinal and/or systematic inflammation is even more limited in humans, as it lacks evidence that correlates gut microbiota changes with disease severity or gender variability. Further studies are needed in order to demonstrate the role of fungi and viruses in the microbiome and to validate the hypothesis that microbial alterations observed in patients with Parkinson’s disease are a cause rather than an effect of the disease.

## 5. Conclusions

Parkinson’s disease is a complex, multifactorial disease regulated by both genetic and environmental factors. Gut microbiome is in constant interaction with the host, affecting the function of the intestinal epithelium, immune cells, and neurons, therefore regulating gut–brain interaction. In this review, we discuss the hypothesis that gut microbiota dysbiosis and its metabolites can provoke intestinal inflammation as well as dysregulation of gut-brain communication and the brain barrier, gradually leading to neuroinflammation and neurodegeneration. Further studies will enable novel approaches to the prevention and treatment of neurological diseases.

## Figures and Tables

**Figure 1 biology-12-00093-f001:**
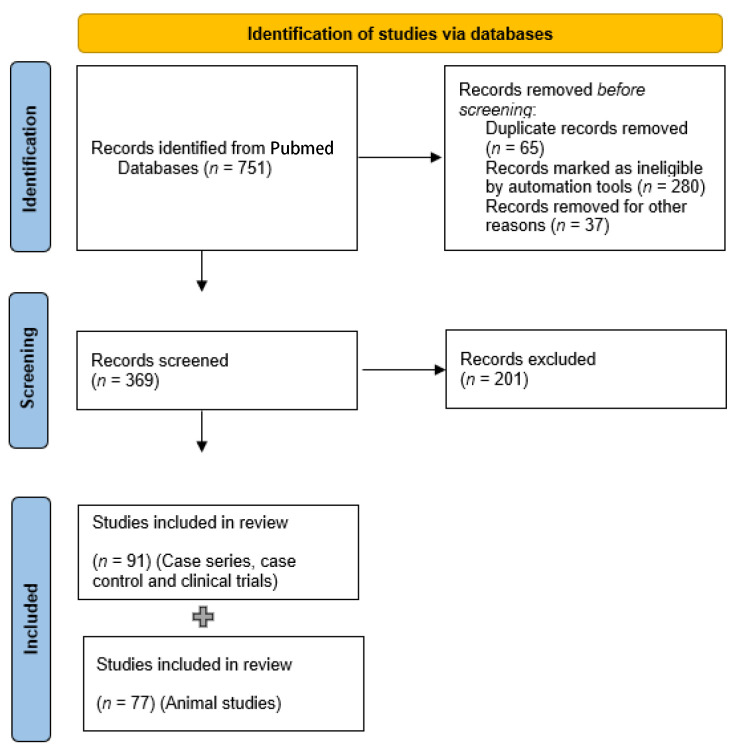
Flow diagram describing the process of identification, screening and final selection of the studies used in this review.

**Figure 2 biology-12-00093-f002:**
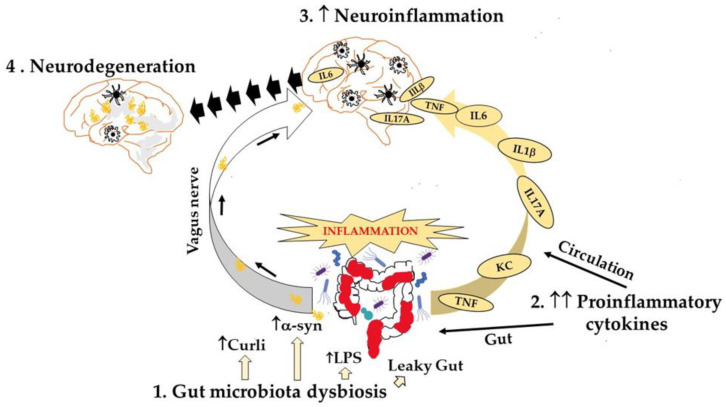
Gut microbiota dysbiosis leads to neurodegeneration. Dysbiosis in the gut can cause (a) inflammation with increased levels of proinflammatory cytokines locally, in circulation, and in brain, (b) induction of a-synuclein aggregates in the gut that via the vagus nerve promote further aggregation in the brain where a-synuclein activates microglia and astrocytes and (c) neuroinflammation that over time leads to neurodegeneration.

**Table 1 biology-12-00093-t001:** Tissue-specific cytokine levels in various experimental animal models of PD and in vitro approaches.

**Chemically-Induced PD Models**				
**Experimental Model**	**Treatment**	**Tissues**	**Cytokine Chemokine**	**Ref.**
MPTP-induced	FMTS	B, C	↓TNF	[73]
	FMT	B	↓TNF	[74]
	FMD	B	↑TNF, IL-1β, IL-17A	[75]
		B, C	↓TNF, IL-6	[76]
	Polymannuronic acid	S	↓IL-6, IL-10	[77]
	TNF inhibitor	CSF, P	↓TNF, IL-6	[72]
	Casein	B, I	↑IL-22, IFN-γ	[78]
	Ceftriaxone	S	↓TNF, IL-6, IL-1β	[79]
	Ceftriaxone	I	↑TNF	[80]
			↑TNF, IL-6, IL-1β,	
			IL-10, IFN-γ, IL-17A,	
			IL-17F, IL-22	
	Chicoric acid	S, B, C	↓TNF, IL-6, IL-1β	[81]
	Cord blood plasma	B, G	↑TNF	[82]
	Vancomycin	C	↓TNF	[83]
	Engineered bacterium	B	↓TNF, IL-6, IL-1β	[84]
	UC-MSCs	S, C	↑TNF, IL-6	[85]
	*Lactobacillus plantarum*	S	↓TNF, IL-6, IL-1β	[71]
			↑IL-10	
	*Lactobacillus plantarum*	B, S	↓TNF, IL-6, IL-1β	[86]
Rotenone-induced	FMT FMT FMT FLZ	B, C, S S B B, C	↓TNF, IL-6, IL-1β ↑TNF, IL-6 ↓TNF, IL-1β ↑TNF, IL-6, IL-1β	[87] [88] [89] [90]
6-OHDA	Antibiotics	B	↓TNF, IL-6, IL-1β	[70]
	hUBC+P	B, G	↓TNF	[91]
	*Helicobacter suis*	P, S	↑ IL-1β, KC	[92]
			⇔ IL-6, IL-10, TNF, IL17A	
	TPG	S	↓TNF, IL-2, IL-1β, IL-4, IL-6	[93]
	L-DOPA	S	↑TNF	[94]
**Transgenic Animal Models**				
**Experimental Model**	**Tissues**	**Cytokine/Chemokine**		**Ref.**
AAV-α-synuclein	I	⇔ TNF, IL-6, IL-1β		[95]
MitoPark	I, C	↑TNF		[96]
*Proteus mirabilis*	C	↑TNF		[97]
(LRRK2*R1441G)135Cjli/J mice	S B	↑IL-17A ↑TNF, IL-1β		[98]
Thy1-αSyn mice	S	↑TNF, IL-1b, IL-6, IL-10		[99]
SNCA-TG mice	F C B	↓IFN-γ, IL-12p70,↑MCP-1 IL-8 signalling, TH-1 responses, ↓TNF, IL-6		[69] [68]
**In Vitro**				
**Cell line**	**Treatment**	**Cytokine/Chemokine**		**Ref.**
Caco-2	With PD, human colonic extracts	↑TNF, CXCL10, ↓IL-6, IL-10		[64]

Abbreviations: FMT: fecal microbiota transplantation, FMD: fasting-mimicking diet, 6-OHDA: 6-hydroxydopamine, TPG: Tianqi Pingchan Granule, DSS: Dextran Sodium Sulfate, MCP-1: Monocyte chemoattractant protein-1, B: brain, C: colon, I: ileum, G: gut, S: serum, P: plasma, CSF: cerebrospinal fluid, F: feces, UC-MSCs: umbilical cord mesenchymal stem cells.

## Data Availability

Not applicable.

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
