# Peer review of "Parkinson’s Disease, It Takes Guts: The Correlation between Intestinal Microbiome and Cytokine Network with Neurodegeneration"

_biology, 2023, doi:10.3390/biology12010093_

Round 1

Reviewer 1 Report

The authors reviewed correlation between intestinal microbiome and cytokine network in Parkinson's disease. I have the following comments:

1. In Figure 1, some numbers for excluded cases are missing

2. Section 3.2 is a bit too short, please elaborate more.

Reviewer 2 Report

in the title and abstract:

is this a narrative review? clarify in the title and abstract.

in the methods:

1. research term should include the vermiform appendix and its postulated role in PD and references added.

2. add a sentence as to which authors performed the extraction of the data.

in the results:

1. use italics where appropriate to indicate bacterial species (see first paragraph section 3.1).  Also check other not used italics (e.g., in vitro).

2. Re table 1:  in the treatment column add additional information such as in those studies reporting FMT add number applications the studies reported and the dose of probiotics used providing a clinical perspective to the study as well as for other treatments (e.g., antibiotics).

Also table 1 appears cumbersome, does this table present human and animal data? Clarify and simplify.

3. In certain parts the results should be presented in the past tense as per usual in scientific reports (see first sentence in section 3.2.2.). Check past tense use throughout the document.

in the discussion:

the discussion should begin with what this study has found and then go on to discuss relevance to the cited literature.

Reviewer 3 Report

The present work is a frankly very poorly thought out and executed systematic review. The questions it tries to solve are too many and it should focus on a specific one. It is not clear throughout the paper which question it is trying to answer. The selected studies are too many and there is not a good description of the findings in the results where only 4 or 5 are mentioned. The authors should focus on a specific type of studies. The discussion is very limited, repeating aspects already described in the introduction. without going into minor aspects, of which there are many, the present work has to be restated with a specific objective, with a good methodical description of the results and with a discussion that includes the main findings, limitations, approaches, hypotheses, etc.